# Youthful systemic milieu in younger recipients alleviates acute kidney injury via attenuating apoptosis and oxidative stress in a rat kidney transplantation model

Chengjun Yu[1,2], Jie Zhang[1,2], Jun Pei[1,2], Shengde Wu[1,2], Sheng Wen[1,2☯*], Yi Hua[1,2☯*], Guanghui Wei[1,2]

1 Department of Urology Children's Hospital of Chongqing Medical University, National Clinical Research Center for Child Health and Disorders, Ministry of Education Key Laboratory of Child Development and Disorders, Chongqing, China, 2 Chongqing Key Laboratory of Structural Birth Defect and Reconstruction, Chongqing, China

☯ Yi Hua and Sheng Wen contribute equally to this work and share corresponding author.
* huayi_workmail@hospital.cqmu.edu.cn (YH); ws851221@163.com (SW)

## Abstract

### Purpose

To evaluate whether acute kidney injury (AKI) differs between young and adult recipients and to explore the possible underlying mechanisms.

### Methods

In an allogenic heterotopic rat kidney transplantation model, we evaluated the renal function, histological damage, oxidative stress injury, apoptosis, immune cell infiltration, and signaling pathways involved in young and adult recipients based on transcriptomics, proteomics, machine learning analyses, and experimental validation. We also identified and validated the possible hub immune-related genes and serum cytokines with the intersection of the maximal clique centrality method, betweenness centrality, and random forest algorithms.

### Results

Younger recipients had lower levels of renal histological damage injury score, oxidative stress injury, and apoptosis. The cytokine profile was different in young and adult recipients. Cytokines and cytokine-mediated pathways involved in oxidative stress, apoptosis, proliferation, and immune cell infiltration were closely correlated with differentially expressed genes. In addition, cytokines mediated immune response participating in antioxidative, anti-apoptosis, and regeneration processes.

**Data availability statement:** All RNA-sequence data has been uploaded to the GEO database (accession number GSE226680), and all relevant data are within the manuscript and its Supporting Information files.

**Funding:** This research was funded by Chongqing Natural Science Foundation Innovation and Development Joint Fund Project (No. CSTB2024NSCQ-LZX0072), Program for Youth Innovation in Future Medicine, Chongqing Medical University (W0201), and The Joint Medical Research Project of Chongqing Science and Technology Bureau and Health Commission (No.2024MSXM002).

**Competing interests:** The authors have declared no competing interests exist.

**Abbreviations:** AKI, acute kidney injury; CKD, chronic kidney disease; ESRD, end-stage renal disease; KTx, kidney transplantation; PPI, protein–protein interaction; KEGG, Kyoto encyclopedia of genes and genomes; GO, gene ontology; GSVA, gene set variation analysis; GSEA, gene set enrichment analysis; DEGs, differentially expressed genes; Socs3, suppressor of cytokine signaling 3; Met, mesenchymal-epithelial transition factor.

## Conclusions

Younger systemic internal milieu in young recipients alleviated AKI after kidney transplantation by ameliorating oxidative stress and apoptosis via immune response regulation. Cytokines may play a protective role in AKI after kidney transplantation. The use of cytokines in transplantation deserves further experimental evaluation and clinical considerations.

---

## 1 Introduction

Chronic kidney disease (CKD) is a worldwide health problem with about 690 million all-stage CKD patients [1], and its prevalence is still increasing globally [2]. Patients with CKD have an increased risk for cardiovascular disease and mortality [3], and about 2% of CKD patients suffer from end-stage renal disease (ESRD) [1]. Renal replacement dialysis therapy and kidney transplantation (KTx) are the only choices for life-supporting for ESRD patients. KTx is the optimal therapy with a lower risk of cardiovascular disease and infection, and a higher level of body health, life quality, and social functions [4]. However, acute kidney injury (AKI) is the most critical and early factor affecting the function and long-term survival of graft after transplantation, how to alleviate AKI after KTx is still a big problem for nephrologists and urologists.

Compared to adult interventions, pediatric kidney transplantation is a much more technical challenging and complicated postoperative management procedure because children are under continuous growth and development [5]. Interestingly, although pediatric KTx has a high prevalence of thrombosis, delayed graft function, and operative failure, the patient and graft survival rates in the pediatric population are still much higher than those in adults, according to the latest Organ Procurement and Transplantation Network/ Scientific Registry of Transplant Recipients (OPTN/ SRTR) report [6]. In addition, we hypotheses that the youthful systemic milieu in children may promote graft and patient survival in some way.

In recent years, significant progress has been made in studying the impact of the youthful systemic milieu on the rejuvenation and regeneration of old organs. In experimental animal studies, the youthful systemic milieu restored the proliferative potential of hepatocytes and the regenerative capacity of aged muscle satellite cells [7], increased fracture repair [8], stimulated the regeneration of pancreatic islet B cells [9], restored the hematopoietic function [10], and promoted renal tubular repair in aged kidneys [11]. It also reversed myelin regeneration in aged rats [12], promoted neurogenesis, improved cognitive function [13], and even reversed myocardial hypertrophy in aged rats [14]. Interestingly, it was further found that the rejuvenation and regeneration of old organs were not caused by the migration of stem cells from young tissues to old organs but by the activation of the proliferation and regeneration potential of aged stem cells by soluble stimulating factors from young body fluids [8,12]. These results indicated that stem cells from old tissues still retained their proliferative and differentiative potential, which provided us with new strategies to mitigate injury and promote regeneration and repair.

In kidneys, Liu *et al.* found that a youthful systemic milieu ameliorated renal ischemia-reperfusion injury in old rats via decreasing oxidative stress, apoptosis, and inflammation in a parabiosis model [15]. At the same time, the exact mechanisms are still unclear. Inspired by this finding, we established a heterotopic rat kidney transplantation model with young (6 weeks old) and adult (12 weeks old) recipients receiving similar donor kidneys to explore the impact of a young systemic environment on AKI and the underlying mechanisms. In the present study, based on multi-omics and a machine learning algorithm, we identified and validated hub regulatory genes and important biological processes involving AKI in allografts 24 h after KTx.

## 2 Materials and methods

### 2.1 Animals

The Ethics Committee of Children's Hospital of Chongqing Medical University approved the study animal use (protocol code CHCMU-IACUC20220429006), Chongqing, China. All procedures regarding animals were in accordance with ARRIVE guidelines (https://arriveguidelines.org) and relevant regulations. The 11-week-old male Sprague-Dawley (SD) rats were purchased from the Experimental Animal Center of Chongqing Medical University (SYXK[YU] 2022−0016, Chongqing, China), and 5- or 11-week-old male Wistar rats were purchased from the Beijing Huafukang Biotechnology Company limited (SCXK[JING] 2019−0008, Beijing, China). All rats were kept in a specific pathogen free environment with a 12-h light/dark cycle at a humidity of 50–60% and a temperature of 23–27°C. All rats have free access to water and food.

### 2.2 Kidney transplantation (KTx), sample harvest, and groups

The rat kidney transplantation process and sample harvest method were described in detail as our previous publication [16,17]. Briefly, rats were anesthetized with pentobarbital at 45 mg/kg via intraperitoneal injection (i.p.) and maintained with isoflurane at 2%, after that, bilaterally nephrectomized Wistar recipients receive male SD left kidneys with an end-to-side anastomosis, and urinary tract reconstruction. At last, a single intramuscular injection of penicillin (1000U/10g) and buprenorphine (0.05 mg/kg) was administered after closing the abdomen. While in sample harvesting, rats were sacrificed with superfluous intraperitoneal pentobarbital injection.

As rats reach sexual maturity at 6–8 weeks and physical maturity at 8–10 weeks, we chose 6- and 12-week-old rats as recipients to mimic pediatric/young KTx and adult KTx, respectively. Briefly, 12-week SD rat donor kidneys were heterotopically transplanted into 6-week Wistar rats (young recipients, Y group) and 12-week Wistar rats (adult recipients, A group).

### 2.3 Histopathological examination for acute tubular necrosis score

Kidney tissues fixed with 4% PFA were embedded in 100% paraffin. The kidney tissues were serially sliced into 4-μm sections. Sections were deparaffinized, rehydrated, and then stained with hematoxylin and eosin (H&E) as previously described [18], or stained with periodic acid-Schiff (PAS) reagent (C0142M, Beyotime, China) according to standard protocols.

Histological damage was assessed in a blinded manner. We quantified the severity of acute tubular necrosis via counting the percentage of tubules displayed cell necrosis, loss of brush borders, protein casts, cast formation, and tubular dilatation in the corticomedullary junction area. The grades was as follows: 0 = none; 1 = ≤10%; 2 = 11–25%; 3 = 26–45%; 4 = 46–75%; and 5 => 76% [11,19]. Approximately 27 randomly chosen high-power fields (HPFs, 400×) per group (3 HPFs per slide, three slides per animal, and three animals per group) were evaluated. Images were observed and captured with a Nikon microscope (Eclipse, Nikon, Japan).

### 2.4 Blood urea nitrogen and Serum creatinine analysis

The serum urea nitrogen and serum creatinine levels in recipients were measured using an autoanalyzer (Cobas C701, Roche, Basel, Switzerland).

## 2.5 RNA sequence *(RNA-seq)*

For RNA sequencing, six renal graft samples were randomly selected from Y and A groups (3 for each). Total RNA was extracted using a Trizol reagent (Thermo Fisher, 15596018, USA) according to the standard manufacturer's protocol. Briefly, after total RNA was extracted, mRNA was purified from total RNA (5 mg) using Dynabeads Oligo (Thermo Fisher, CA, USA) with two rounds of purification. Following purification, the mRNA was enriched and fragmented into short fragments using a fragmentation buffer and reversely transcribed into cDNA by SuperScriptTM II Reverse Transcriptase (Invitrogen, 1896649, USA). At last, we performed the 2 × 150 bp paired-end sequencing (PE150) on an Illumina Novase-qTM 6000 (LC-Bio Technology CO., Ltd., Hangzhou, China) following the recommended protocol. All initial data has been uploaded to the GEO database (accession number GSE226680).

## 2.6 Bioinformatics analysis

**2.6.1 Differentially expressed genes.** Gene differential expression analysis was performed by DESeq2 software between two groups (A vs. Y) [20]. The genes expression profile with a P-value less than 0.05 and absolute fold change ≥1 were considered differentially expressed genes (DEGs) [21]. In addition, the DEGs related to the immune response in different KTx recipients were obtained by intersecting the DEGs with immune-related genes downloaded from the Immunology Database and Analysis Portal (ImmPort; https://www.immport.org/home) [22], which contains 1,793 immune-related genes [21].

**2.6.2 Gene ontology *(GO)* enrichment analysis.** Functional annotation of the DEGs in GO enrichment, including the biological process, molecular function, and cellular component, is a common method for performing functional enrichment analyses of genes. GO can recognize the main biological functions that DEGs exercise. All DEGs were mapped to GO terms in the Gene Ontology database (http://www.geneontology.org/) and visualized using the OmicStudio tools at: https://www.omicstudio.cn/tool. GO terms with a P-value ≤0.05 were regarded as significantly enriched terms.

Bioinformatic analysis was performed using the OmicStudio tools at https://www.omicstudio.cn/tool.

**2.6.3 Kyoto encyclopedia of genes and genomes *(KEGG)* enrichment analysis.** Pathway enrichment analysis was performed using the major public pathway-related well-known database KEGG [23]. Significantly enriched signal transduction pathways in calculated DEGs were identified by comparing them with the whole genome background. Pathways with a P-value less than 0.05 were regarded as significantly enriched.

**2.6.4 Gene set enrichment *(GSEA)* analysis.** We performed gene set enrichment analysis using software GSEA (v4.1.0, Broad Institute, Inc, USA) and Molecular Signatures Database (MSigDB) to identify whether a set of genes in specific GO terms, KEGG pathways, Reactome shows significant differences between the two groups. Briefly, we input the gene expression matrix and rank genes by signal-to-noise normalization method. Enrichment scores p-valuable were calculated in the default algorithm with 1,000 permutations. Gene set size filters were set at a minimum of 5 and a maximum of 5,000. GO terms, KEGG pathways, and Reactome meeting this condition with |normalized enrichment score (NES)| > 1, nominal P-value <0.05, and FDR q-value <0. 25 were considered to be different in the two groups.

To identify the immune profile differences between the A and Y groups, we conducted a GSEA based on the immune background. In addition, to assess the potential difference in biological functions and pathways between the highly (≥ 50%) and lowly expressed (< 50%) hub genes (mainly *Socs3* and *Met*), we downloaded the "c5.go.bp.v7.4.symbols.gmt" gene set from MSigDB (http://www.gsea-msigdb.org/gsea/downloads.jsp) and performed the single-gene GSEA to explore pathways and possible mechanisms [24].

**2.6.5 Network construction and hub gene identification.** The Search Tool for Retrieval of Interacting Genes/Proteins (STRING) database (http://string.org) was used to develop a protein–protein interaction (PPI) network for immune-related DEGs and cytokines. Cytoscape software (Version 3.9.1, Cytoscape Consortium, USA) was used to optimize the visualization of networks. Systematic analysis of hub genes and cytokines is vital to understand how these proteins reciprocally work in biological systems, respond to stress, regulate signaling transduction and metabolism in

specific physiological states, and explore the functional connections between proteins. The top 10 hub genes of these interactions were identified using the Cytoscape cytoHubba plug-in according to the "Maximal Clique Centrality (MCC)" method [25], cytoNCA plug-in according to the "Betweenness Centrality (BC)" method, and a machine learning-based random forest method. And then, we took the intersection of immune-related DEGs calculated by these three different algorithms.

### 2.6.6 Gene set variation analysis (GSVA).

To assess the potential differences in biological functions and signaling pathways between the highly (≥ 50%) and lowly expressed (< 50%) hub-gene subgroups, we used GSVA to explore significantly enriched signaling pathways. GSVA is an enrichment method used to analyze pathway activity variations in a simple population unsupervised manner. Briefly, we calculated every sample's enrichment score in GSVA (v1.40.1) by gene expression profile according to the method described by Hänzelmann *et al.* [26], obtained the enrichment score matrix, and then downloaded the "c5.go.v7.4.symbols.gmt" gene set from MSigDB [24], to explore biological functions and possible molecular mechanism. Variance analysis of GSVA results was conducted using the "limma" package in R to uncover gene sets that were significantly different [27]. Gene set size filters were set at a minimum of 5 and a maximum of 5,000. Plots were visualized by the Sangerbox online tool [28], at: http://sangerbox.com/home.html. Significant biological functions and pathways variations were defined as a P-value < 0.05.

### 2.6.7 Immune cell infiltration analysis.

Immune cell infiltration is the base of the immune response. CIBERSORT is a computational algorithm for quantifying cell fractions based on gene expression profiles. In this study, 22 types of immune cells in kidney transplantation samples were evaluated and calculated by R package "IOBR" using the CIBERSORT algorithm [29,30], which was based on the single sample gene set enrichment analysis (ssGSEA). In addition, the association between hub immune-related genes and immune infiltrating was evaluated by Spearman's correlation analysis.

## 2.7 Measurements of malondialdehyde levels (MDA), total superoxide dismutase activity (T-SOD), glutathione peroxidase activity (GSH-Px), and protein carbonyl content (PC)

Kidney samples were homogenized on ice. Then, the tissue supernatant was extracted after centrifuging at 4°C. Malondi-aldehyde levels, total superoxide dismutase activity, glutathione peroxidase activity (Nanjing Jiancheng Biochemistry Co., Nanjing, China), and protein carbonyl (BC1275, Solarbio, China) content were determined with commercial kits according to the manufacturer's instructions. Optimal density values were determined by the Gen5 Microplate Reader and Imager Software (Version 3.11, BioTek Instruments, Inc., USA).

## 2.8 TUNEL analysis

A One-Step TUNEL Assay Kit (Andy Fluor™ 594, A051, USA) was used for the graft kidney TUNEL assay according to the manufacturer's instructions. Immunofluorescence imaging was performed using an A1R confocal microscopy (Nikon, Tokyo, Japan).

## 2.9 5-Ethynyl-2′-deoxyuridine assays

The 5-Ethynyl-2′-deoxyuridine (EdU) staining method of graft kidney was performed according to the standard protocol as previously described [16], by the iClick™ EdU Andy Fluor 555 Imaging Kit (A008, Andy Fluor, USA). The recipient rats were intraperitoneally injected with 5 μg/g EdU after KTx surgery. The slides with frozen kidney sections were fixed with 4% PFA for 15 min at room temperature, permeabilized with 0.5% Triton X-100 (T8200, Solarbio, China) in PBS for 20 min, rinsed 3 times for 10 min each, and incubated with iClick reaction cocktail for 30 min in the dark at room temperature. Nuclear counterstaining was performed using Hoechst 33342, and then the samples were mounted with Antifade Mount-ing Medium (P0126, Beyotime, China) after rinsing 2 times in phosphate buffer saline (PBS). Images were obtained by an A1R confocal microscopy.

## 2.10 Protein extract and western blotting

Proteins were extracted from adult and young Wistar recipient rats' kidneys with a radioimmunoprecipitation (RIPA) lysis buffer (P0013B, Beyotime Biotech, China) supplemented with 1% phenylmethanesulfonyl fluoride (ST506, Beyotime Biotech, China). Protein concentrations were measured by the bicinchoninic acid (BCA) method according to the standard procedures (Thermo Scientific, USA). Before proteins were loaded onto sodium dodecyl sulfate-polyacrylamide gel electrophoresis (SDS-PAGE) gels, proteins were mixed with 5×SDS-PAGE loading buffer and then boiled for 10 min.

Total proteins (20 μg for each well) were separated by 10% or 12.5% SDS-PAGE and then transferred to polyvinylidene fluoride (PVDF) membranes (Millipore, USA). After blocking with quick blot blocking buffer (P30500, NCM, China) for 10 min, the membranes were incubated with primary antibodies as cleaved caspase 3 (9661, Cell Signaling Technology, USA), Bax (50599–2-Ig, Proteintech, China), Bcl-2 (381702, ZENBIO, China), Socs3 (A0769, Abclonal, China), Met (R24961, ZENBIO, China), and GAPDH (200306, ZENBIO, China) overnight at 4°C. GAPDH was regarded as a loading control. Following three washes with Tris-buffered saline plus Tween-20 (TBST), the membranes were incubated with goat-anti-rabbit secondary antibody conjugated with horseradish peroxidase (HRP) for 1 h at room temperature. Chemiluminescent reactions (Millipore, USA) were used to detect the membrane bands, and signals were visualized by the ChemiDoc™ Touch Imaging System (Bio-Rad, USA). Integrated relative densities of individual bands were quantified using the Image Lab software (Version 6.0.1, Bio-Rad Laboratories, Inc., USA).

## 2.11 Quantitative polymerase chain reaction (qPCR)

Detailed experimental protocols were described previously [31]. Briefly, kidney tissue RNA was isolated from frozen tissues stored at –80 °C, RNA was purified and cDNA generated, and then, cDNA was amplified using 40 cycles with the following condition: 95 °C for 30 s, 95 °C for 15 s, and 60 °C for 30 s. The expression of target protein was based on the formula $2^{-\Delta\Delta Ct}$, and the primers used are listed in S1 Table.

## 2.12 Transmission Electron Microscopy

Kidney samples were prefixed with 3% glutaraldehyde, then postfixed in 1% osmium tetroxide, dehydrated in series acetone, and embedded in Epox 812. First, semithin sections were stained with methylene blue for optical localization, and then, ultrathin sections were cut with a diamond knife (EM UC7, Leica, Germany) and stained with uranyl acetate and lead citrate. Finally, sections were examined with JEM-1400-FLASH Transmission Electron Microscope (JEOL, Japan).

## 2.13 Cytokine array and analysis

The detailed method of cytokine array was as previously described [16]. More than 500 serum proteins were tested in young and adult recipients using the L-Series Label-Based Rat Antibody Array 3 (RayBiotech, USA). A cytokine with fold change (FC) of ≥1.2 or <0.83 and a P-value <0.05 was considered statistically differentially expressed [32,33]. For differentially expressed cytokines, the GO and KEGG enrichment analysis, PPI network, and random forest were conducted.

## 2.14 Protein–protein docking

Proteins are central to many processes in living cells while determining the 3D structure and interaction between proteins is a critical step to understanding the physical mechanisms of these processes. We conducted the protein 3D structure and protein-protein docking with an online tool at https://www.dockeasy.cn.

## 2.15 Statistical analysis

Data were presented as mean±standard error of the mean (SEM). Differences between groups were compared with Student's t-tests using GraphPad Prism (version 9.0.0, GraphPad Software Inc., USA). The analysis of correlations was performed using

Spearman's correlation method. Each experiment was repeated at least three times. R software and attached R packages were used for all bioinformatics analyses. A *P*-value less than 0.05 was considered statistically significant.

## 3 Results

### 3.1 Youthful systemic milieu in young recipients protected renal function and alleviated histological damage after allogenic kidney transplantation

In this study, 12-week-old (adult) SD rats' donor kidneys were heterotopically transplanted into 6-week-old (young) and 12-week-old (adult) Wistar rats to mimic pediatric and adult recipients to explore the impact of youthful systemic milieu on allografts injury. The study design diagram and kidney transplantation procedures are exhibited in Fig 1A and B.

Twenty-four hours after allogenic kidney transplantation, compared with the Y group, the levels of BUN and serum creatinine (Cr) were significantly higher in the A group (Fig 1C and D). Twenty-four hours after KTx, compared with the Y group, the adult recipients showed more serious tubular cell necrosis, a wide loss of brush border, cast formation, tubular lumens dilation, and even protein cast formation (Fig 1E). The tubular injury score and percent of necrotic tubules were also higher in the A group (Fig 1F and G). These results indicated that the younger systemic internal environment protected renal function and alleviated renal histological injury in some way after allogenic KTx.

### 3.2 RNA sequencing and enrichment analyses revealed that immune response, apoptosis process, and oxidative phosphorylation levels were differentially expressed

To explore why young recipients' younger internal environment alleviated the renal damage and underlying mechanisms, we randomly collected three kidney samples from each of Y and A groups for RNA sequencing analysis. Analysis of RNA differential expression identified 461 DEGs between A and Y groups, including 219 upregulated genes and 242 downregulated genes (Fig 2A). The distribution of obtained DEGs is depicted in a volcano plot (Fig 2A). Hierarchical clustering of DEGs was conducted and depicted as an expression heat map (Fig 2B), which showed a different expression pattern between the two groups.

We performed GO and KEGG pathway enrichment analyses for a more in-depth view of identified DEGs in biological functions and signaling pathways. The GO results demonstrated that DEGs between A and Y groups were mainly associated with the regulation of cytokine production involved in immune response, immature T cell proliferation, epithelial cell apoptosis, T cell tolerance induction, T cell differentiation, oxygen transport, response to cytokine, cytokine receptor activity, inflammatory response, and oxidative stress (Fig 2C). The KEGG results indicated that the enriched pathways are mainly involved in retinol metabolism, mineral absorption, glycolysis and gluconeogenesis, renin-angiotensin system, PPAR and JAK-STAT signaling pathway (Fig 2D).

One limitation of DEG enrichment analyses is that a minority of key genes with mild expression but exhibiting important regulation functions may be left out due to DEG criteria. Thus, we also conducted a GSEA to investigate the potential functional alternations between young and adult recipients after receiving similar donor kidneys. As expected, GSEA analyses revealed that the oxidative phosphorylation process was downregulated in young recipients with an NES of −3.1039 and an FDR of 0.0 (Fig 2E). However, the T cell receptor signaling pathway, B cell receptor signaling pathway, and apoptosis process were significantly upregulated in adult recipients (Fig 2F–H). These results indicated that the allografts in young and adult recipients might have a different extent of the immune response, apoptosis, and oxidative phosphorylation levels, due to an adult systemic internal milieu (12 weeks) and a younger one (6 weeks). In addition, cytokines may play a pivotal role in these two scenarios.

### 3.3 Young recipients had lower levels of apoptosis and oxidative stress damage compared with adult recipients

To verify the findings based on RNA-seq techniques, we performed experiments to evaluate the oxidative stress levels, apoptosis, and proliferation state in allografts. Twenty-four hours after KTx, compared with adult recipients, kidney extracts

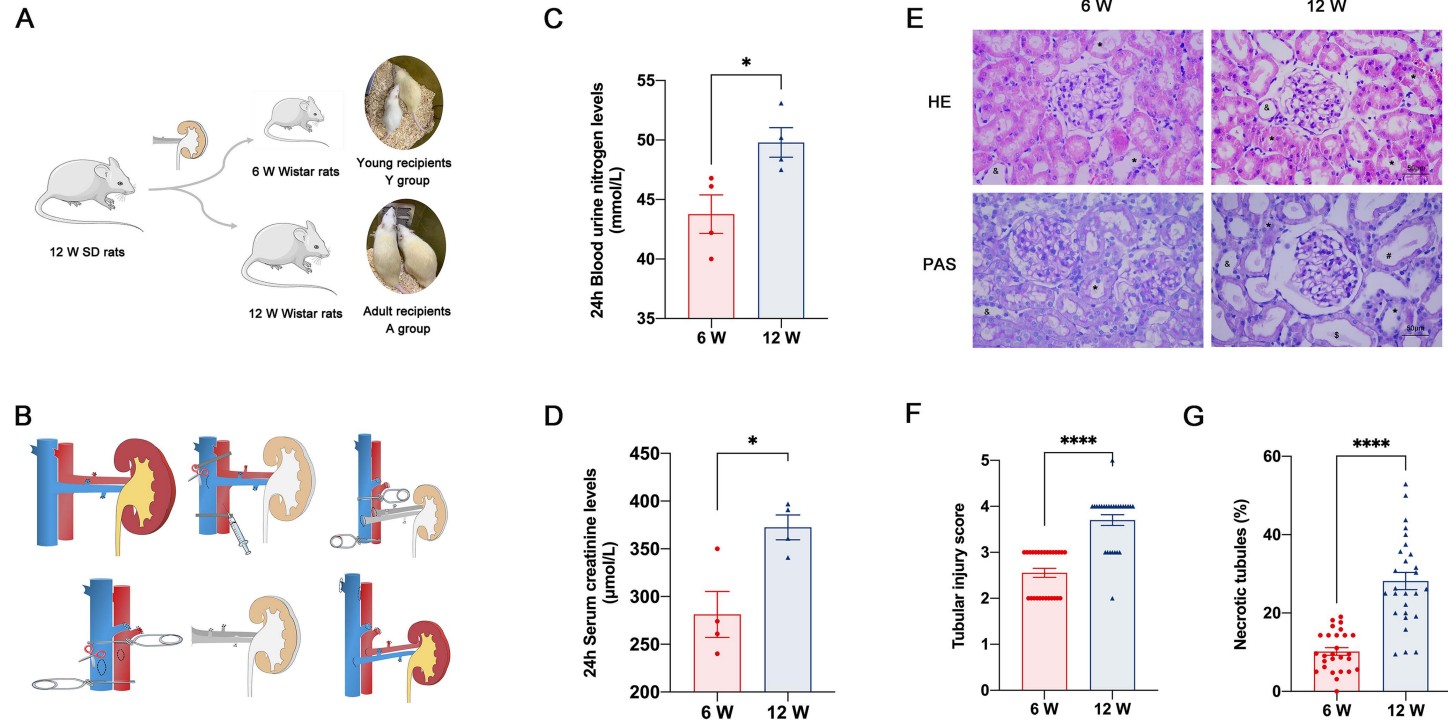

**Fig 1. Youthful systemic internal milieu in young recipients protected renal function and alleviated histological damage.** (A) Study design diagram. (B) Allograft heterotopic kidney transplantation and end-to-side vascular anastomosis procedures. (C) Blood urea nitrogen levels and (D) serum creatinine levels in young (6 W) and adult (12 W) recipients 24 hours after KTx (for C and D, $n = 4$ in each group). (E) Representative photographs of kidney sections from recipients stained with hematoxylin and eosin and periodic acid-Schiff (*indicates necrotic tubulars, #indicates loss of brush borders, &indicates tubular dilation, $indicates protein cast formation; original magnification, 400×, the scale bar represents 50 μm). (F) Renal tubular injury score and (G) percent of necrotic tubulars in Y and A group (for F and G, 3 HPFs per slide, 3 slides per animal, and 3 animals per group, 27 randomly chosen high-power fields in total in each group were evaluated). Values were presented as mean ± SEM. *$P < 0.05$, ****$P < 0.0001$ compared with the A group. HE, hematoxylin and eosin staining, PAS, periodic acid-Schiff staining. Approximately 27 randomly chosen high-power fields (HPFs, 400×) per group (3 HPFs per slide, three slides per animal, and three animals per group) were evaluated.

from young recipients showed lower levels of malondialdehyde and protein carbonyl content, but higher levels of total superoxide dismutase activity and glutathione peroxidase activity (Fig 3A–D). Electron microscopy showed that mitochondria morphology in young recipient kidneys was relatively normal, with a complete membrane compared with the A group, whereas adult recipients showed more swelled mitochondria with broken membranes and dissolved ridges (Fig 3E).

For apoptosis evaluation, the TUNEL-positive tubular cells were significantly lower in the Y group than that in A group (Fig 3F). Moreover, the expression levels of Bax and cleaved caspase 3 were lower in the Y group compared with the A group. At the same time, the anti-apoptosis protein, Bcl-2, was higher in the Y group (Fig 3G and I). In addition, EdU assays revealed that 24 h after allogenic KTx, young recipients had a higher rate of renewed epithelial tubular cells than adult recipients (Fig 3H). These results indicated that the youthful systemic milieu in young recipients decreased the apoptosis and oxidative stress injury and may promote tubular repair after kidney transplantation.

### 3.4 Immune cell infiltration and immune cell state differed in both groups, and cytokines may regulate these processes

Immune-related acute and chronic rejection and even acute kidney injury (AKI) are inevitable biological processes in kidney transplantation. Our previous analyses revealed that young and adult recipients who received similar donor kidneys

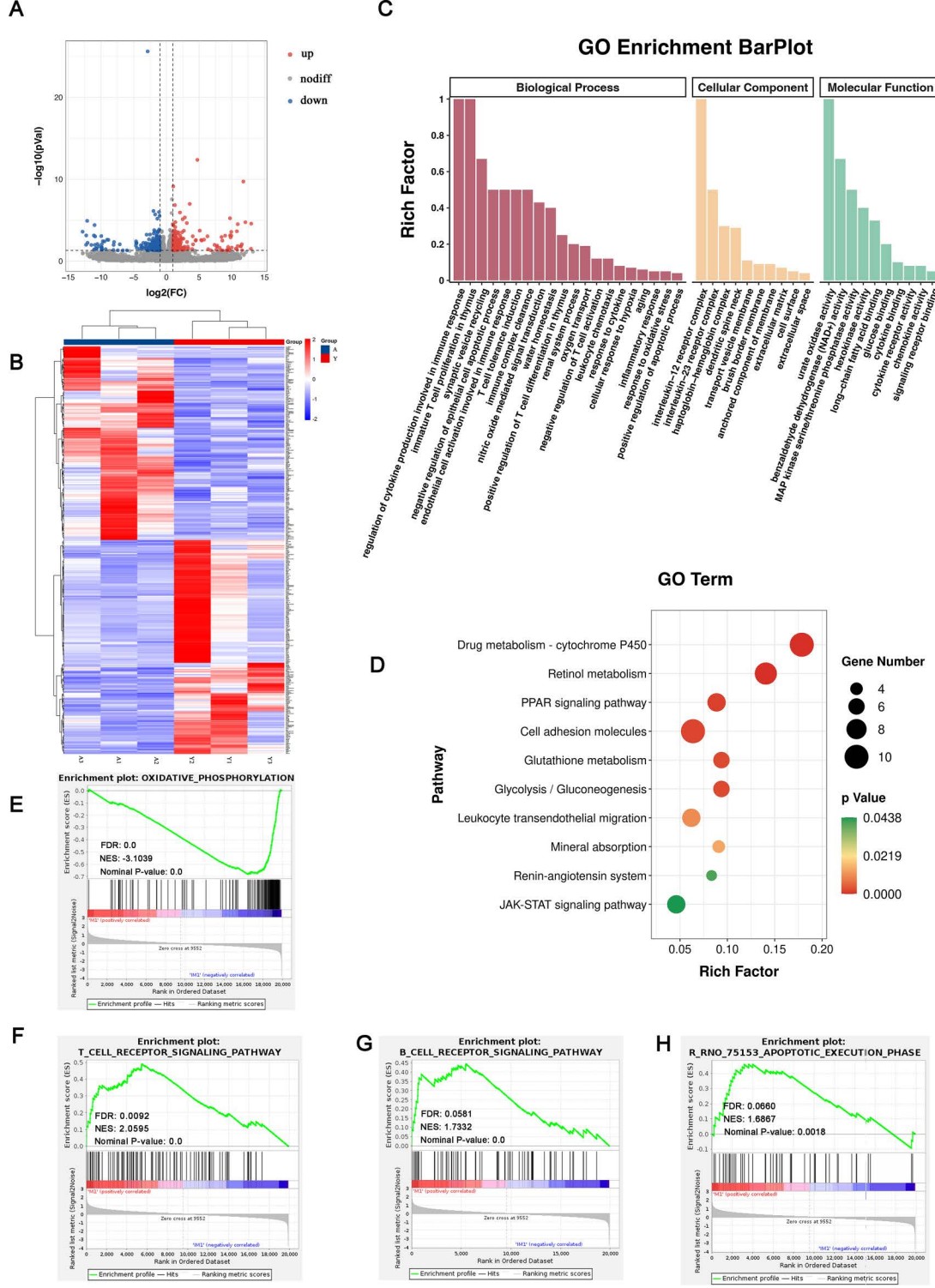

**Fig 2. Transcriptive analysis of allograft kidneys from young and adult recipients.** (A) Volcano plot for DEGs. (B) Heatmap for DEGs between Y and A group. (C) GO enrichment and (D) KEGG enrichment analysis based on DEGs. GSEA analysis on oxidative phosphorylation (E), T cell receptor signaling pathway (F), B cell receptor signaling pathway (G), and apoptotic execution phage (H).

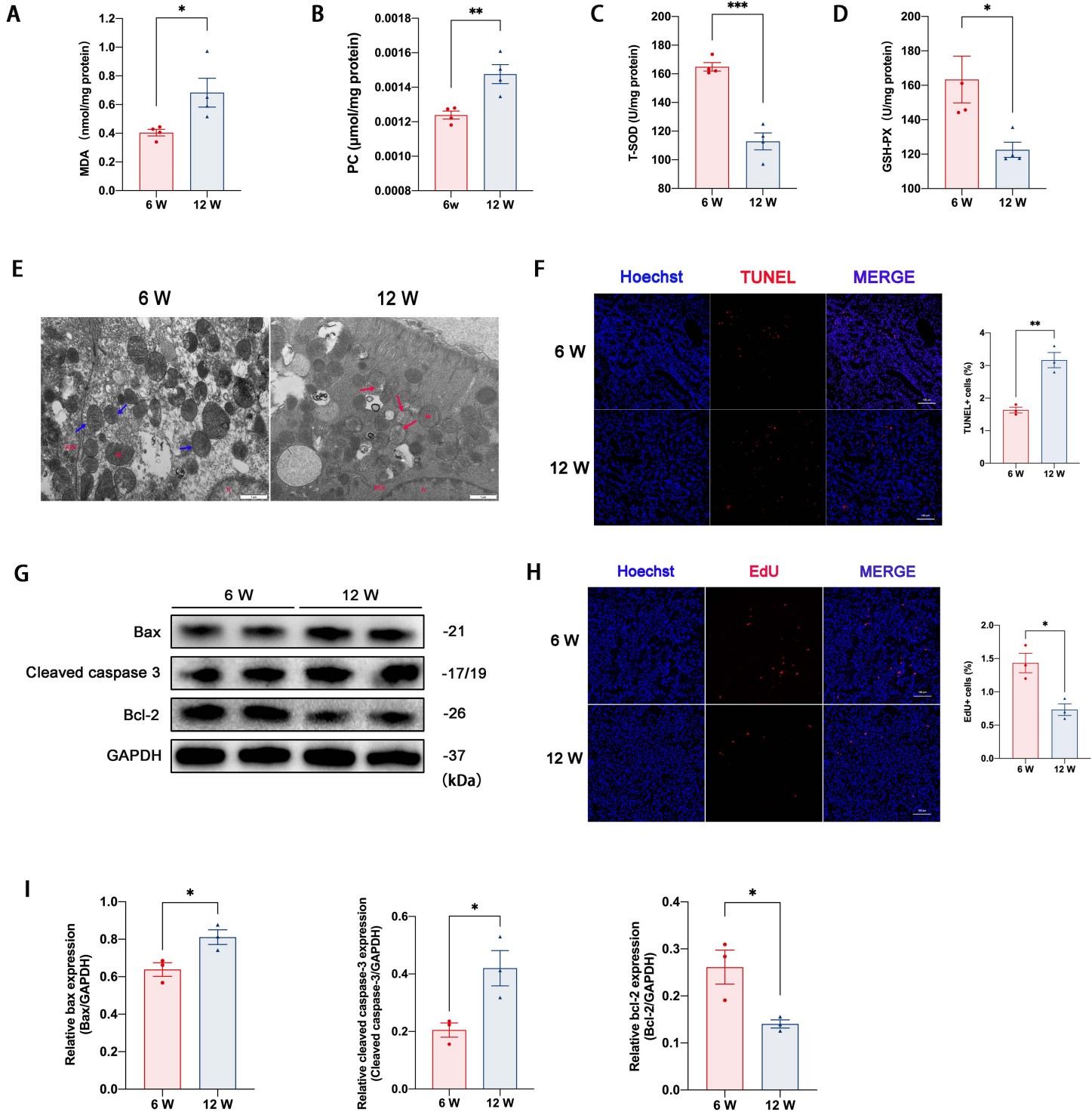

**Fig 3. Youthful systemic milieu in young recipients decreased apoptosis and oxidative stress damage in allografts, and promoted proliferation.** (A–E) Oxidative stress injury evaluation in allografts from Y and A group. (A) Malondialdehyde (MDA) levels, (B) protein carbonyl (PC) content, (C) total superoxide dismutase activity (T-SOD), and (D) glutathione peroxidase activity (GSH-Px) in kidney extracts of allografts (for A–D, $n = 4$ in each group). (E) Representative transmission electron microscopy photographs of mitochondria morphology from Y and A group (Blue arrows denote relatively normal mitochondria morphology with complete membrane, and red arrows denote swelling mitochondria with damaged membrane; N, nuclear,

Mi, mitochondria, RER, rough endoplasmic reticulum; original magnification, 20000×, the scale bar represents 1 μm). (F) Representative images of kidney sections stained by TUNEL assays (original magnification, 200×, the scale bar represents 100 μm). (G) Representative western blotting bands of pro-apoptosis (Bax, cleaved caspase 3) and anti-apoptosis (bcl-2) protein markers. (H) Representative photographs of frozen kidney sections stained by 5-Ethynyl-2′-deoxyuridine (EdU) assays reagents (original magnification, 200×, the scale bar represents 100 μm). (I) Quantitative analyses of the bands desities of bax, cleaved caspase 3, and bcl-2 protein expression ($n=3$). Values were presented as mean±SEM. *$P<0.05$, *$P<0.01$, ***$P<0.001$ compared with the A group.

demonstrated different states of immune response 24 h after KTx. Furthermore, GSEA analyses based on immune background confirmed this conclusion (Fig 4A–C). Furthermore, an internal systemic environment in young and adult recipients affected the B cell and T cell activation, proliferation, and differentiation (Fig 2C, F, G; Fig 4A–C). In addition, adult recipients had higher expression levels of plasma cells, which may also indicate a higher possibility of antibody-mediated rejection. Different immune response states in young and adult recipients might account for the different levels of renal functions and histological damage.

Based on the gene expression profile, we evaluated the infiltration state of 22 types of immune cells in kidney transplantation samples by the CIBERSORT algorithm. The infiltration results were visualized as a bar plot (Fig 4D). The

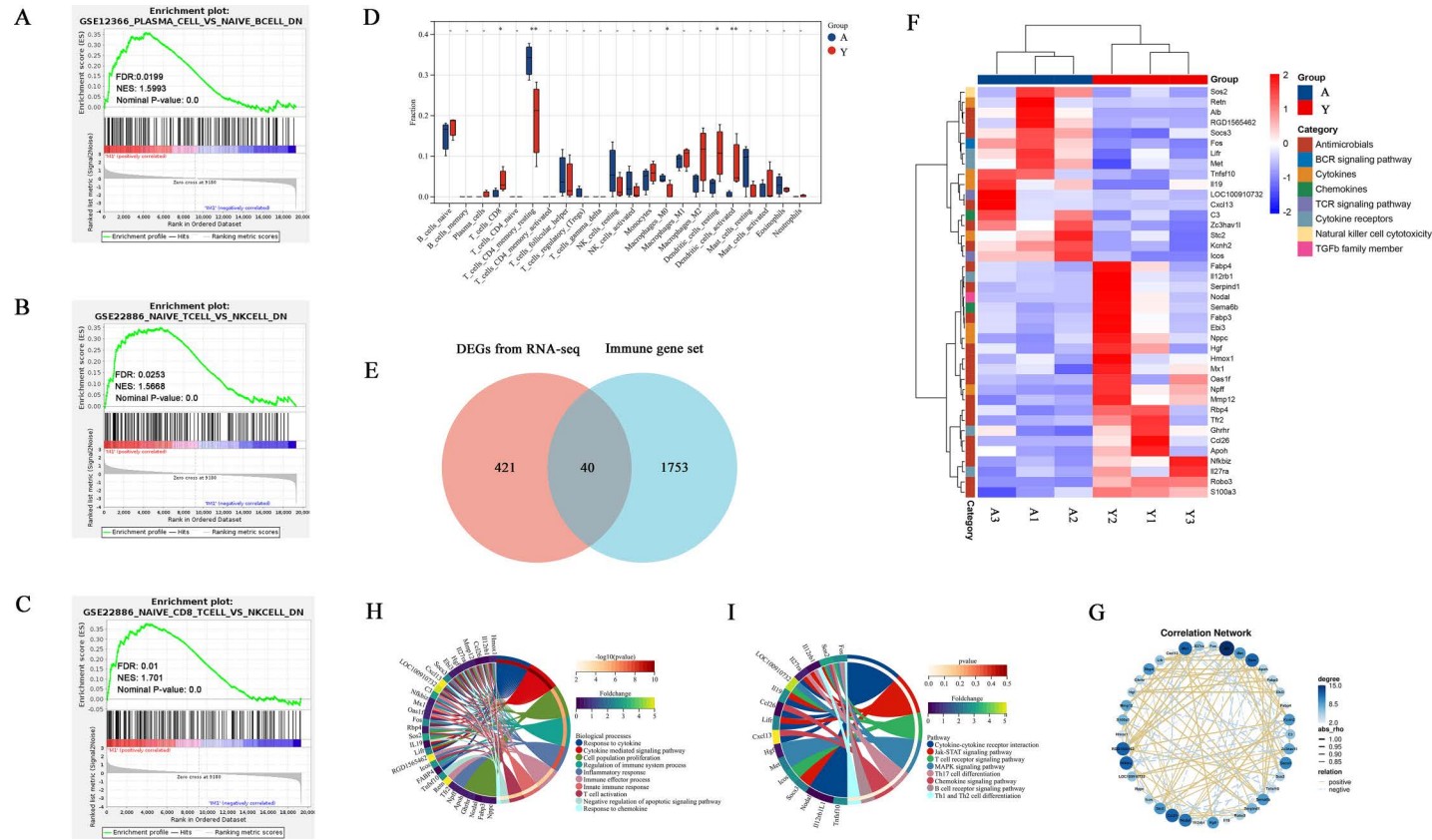

**Fig 4. Immune cell infiltration and immune cell state were different in adult and young recipient kidneys.** (A-C) GSEA analyses based on immune background. (D) The bar plot of 22 types of immune cells infiltration calculated by CIBERSORT algorithm ($n=3$). (E) The intersection of DEGs from RNA-seq and immune gene sets downloaded from Immunology Database and Analysis Portal (ImmPort). (F) The correlation heatmap and (G) correlation network diagram of the identified 40 immune-related DEGs. (H) The GO enrichment and (I) KEGG enrichment analysis of the identified 40 immune-related DEGs.

proportion of CD8+ T cells, resting dendritic cells, and activated dendritic cells in young recipients were significantly higher than those in adult recipients. Meanwhile, CD4+ memory resting T cells and M0 macrophages in young recipients were significantly lower than in adult recipients (Fig 4D).

To further explore immune-related regulatory genes, we downloaded the full immune gene set from the Immunology Database and Analysis Portal (ImmPort), intersected it with DEGs from RNA-seq, and obtained 40 immune-related genes (Fig 4E). Hierarchical clustering of these 40 immune-related DEGs showed a different expression pattern between the two groups (Fig 4F). At the same time, these genes were also closely correlated and analyzed by correlation network diagram (Fig 4G). In addition, we conducted GO and KEGG enrichment analyses to explore vital biological functions and pathways further. GO results revealed that biological processes were significantly enriched, including response to cytokines, cytokine-mediated signaling pathway, cell proliferation, regulation of the immune system, T cell activation, and response to chemokines (Fig 4H). KEGG results indicated that cytokine–cytokine receptor interaction, JAK-STAT signaling pathway, T cell receptor signaling pathway, MAPK signaling pathway, B cell receptor signaling pathway, and chemokine signaling pathway were significantly enriched, which is consistent with GO terms (Fig 4I). These results indicated that differentially expressed cytokines may regulate immune cell activation, proliferation, differentiation, and infiltration, and then regulated immune response in allografts after kidney transplantation.

### 3.5 *Socs3* and *Met* may be hub genes regulating the immune response in different recipients

To further explore differentially expressed regulatory hub genes involved in immune response in young and adult recipients after KTx, we constructed PPI networks and calculated their priority ranks according to BC (Fig 5A), MCC method (Fig 5B), and random forest algorithm (Fig 5C). Intersection of top 10 ranked genes calculated by these three algorithms yielded three hub genes [Suppressor of cytokine signaling 3(*Socs3*), Mesenchymal-epithelial transition factor (*Met*), and Resistin (*Retn*)] (Fig 5D). The respective PPI network of these three genes was depicted in Fig S1B-D. Considering that *Retn* was expressed at a very, very low level in both Y and A group (Fig S1A), we only validated the expression levels of SOCS3 and MET in kidney samples by western blotting (Fig 5E). As expected, SOCS3 and MET were both highly expressed in adult recipients (Fig 5F), consistent with RNA-seq results.

As immune cell infiltration is the base of the immune response, we identified the correlation of these three hub genes with 22 types of immune cells. Not surprisingly, *Socs3*, *Met*, and *Retn* were all closely correlated with immune cell infiltration (Fig 5G and H, Fig S1E), indicating that these hub genes played a vital role in regulating the immune response in kidneys after KTx. Based on the highly and lowly expressed hub-gene subgroups, we performed the single gene GSEA analyses on biologically processed enrichments and KEGG pathways to explore possible mechanisms and signaling pathways. From these results, low expression of *Socs3* was linked to a higher level of oxidative phosphorylation, response to oxygen radicals, and cell cycle checkpoint regulation. In contrast, high expression of *Socs3* was associated with activating selective autophagy and ERBB signaling pathway (Fig 5I and J). Furthermore, low expression of *Met* was linked to elevated levels of oxidative phosphorylation, response to oxygen radical, and enhanced electron transport, while high expression of *Met* may increase the apoptosis process (Fig 5K and L). Although *Retn* was very lowly expressed, highly and lowly expressed Retn may also have similar biological processes (S1F and G Fig).

In addition, we used GSVA to analyze biological process enrichments and pathway activity variations in highly and lowly expressed hub-gene subgroups by gene expression profile, calculated every sample's enrichment score, and obtained the enrichment score matrix for subsequent analyses. The GSVA analyses activated the T cell receptor signaling pathway, apoptosis process, B cell receptor signaling pathway, and mTOR signaling pathways (Fig 5M and N). At the same time, oxidative phosphorylation was downregulated in the highly expressed hub-gene subgroup, consistent with the foregoing analyses and experimental validations. Furthermore, natural killer cell cytokine production, CD4+αβ T cell cytokine production, and other cytokines were also expressed differently. *Socs3* and *Met* are hub genes regulating the immune response involved in kidney transplantation, affecting oxidative stress, apoptosis, and proliferation.

 

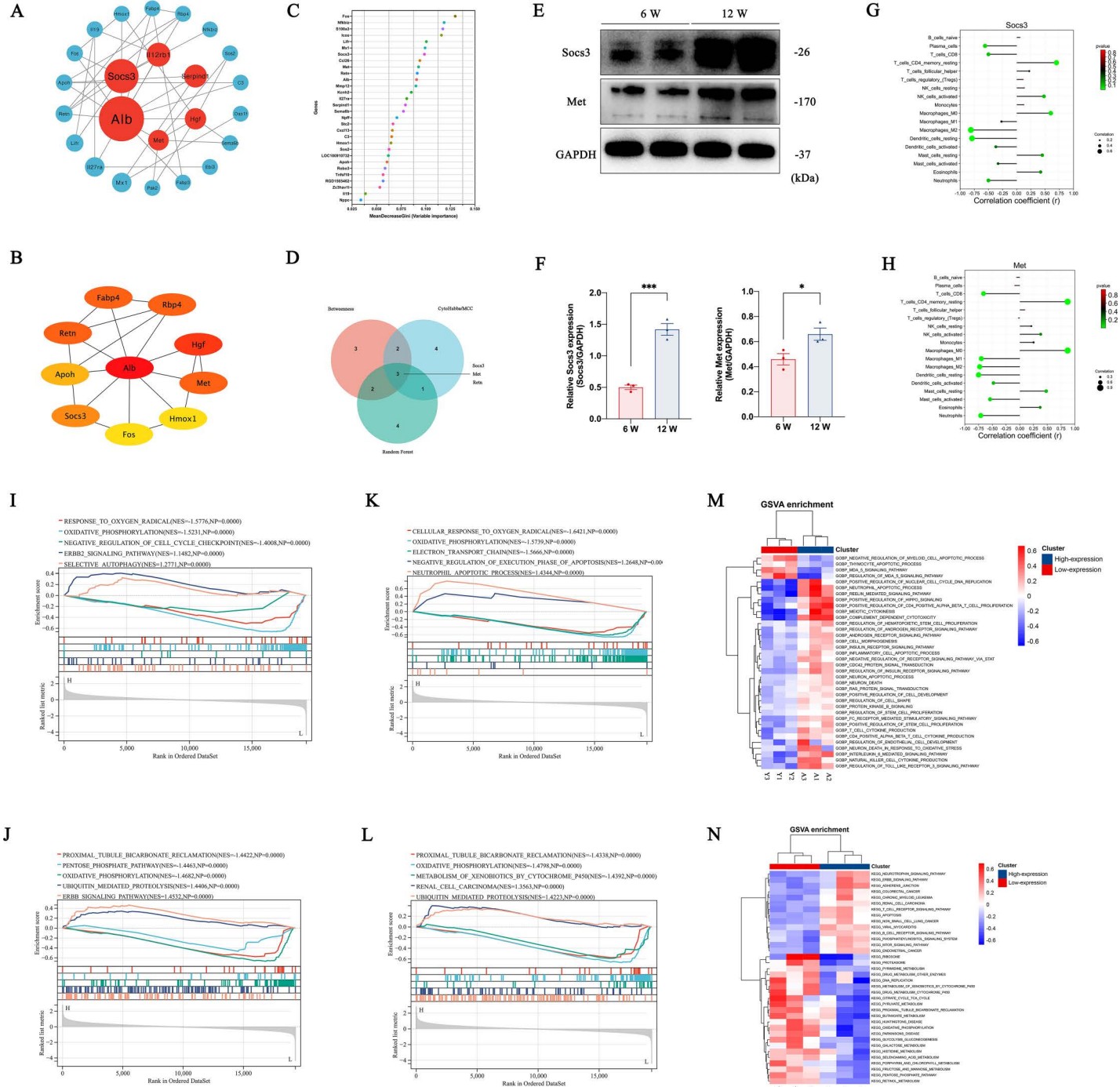

**Fig 5. Identification, validation, and functional analyses of hub immune-related genes.** Network construction of immune-related DEGs according to the Betweenness Centrality method (A) and Maximal Clique Centrality method (B). (C) Identification of top 30 hub genes based on a machine-learning method, random forest. (D) Venn diagram of intersection of top 10 hub genes calculated by Betweenness Centrality, Maximal Clique Centrality, and random forest algorithm. Validation of hub genes (Socs3 and Met) by western blotting (E) and quantitative analyses of the bands density (F). Values were presented as mean ± SEM ($n$ = 3). *$P$ < 0.05, ***$P$ < 0.001. (G) Correlation analysis of Socs3 and 22 types of immune cells. (H) Correlation analysis of Met and 22 types of immune cells. Single gene GSEA analyses of biological processes (I) and KEGG pathways (J) based on high- (≥ 50%) and low-expressed (< 50%) Socs3. Single gene GSEA analyses of biological processes (K) and KEGG pathways (L) based on high- and low-expressed Met. Gene set variation analysis (GSVA) of biological process enrichments (M) and KEGG pathways (N) based on the high- and low-expressed hub genes.

## 3.6 Cytokines played a vital role in regulating the immune response in young and adult recipients after kidney transplantation

Previous analyses revealed that cytokines might play an important role in regulating immune response, affecting oxidative stress, apoptosis, renal function, and proliferation. Thus, we conducted a protein microarray and tested over 500 cytokines in serums from young and adult recipients to identify whether cytokines were differentially expressed in Y and A groups. A total of 500 cytokines were compared between Y and A groups. Analysis of cytokine differential expression identified 143 differentially expressed cytokines, among which 140 types of cytokines were upregulated in the Y group, and only three cytokines (Rheb, TGF-β R1, and WARS) were upregulated in the A group (A versus Y, Fig 6A and B). About 28.6% of total cytokines were differentially expressed in young and adult recipients, and most interestingly, most of these were highly expressed in young rat serum. These results indicated that higher levels of cytokines in the youthful systemic internal milieu might play a protective role in immune response, kidney injury, and tissue repair.

We conducted a GO and KEGG enrichment analysis to further explore the functions and signaling pathways of differentially expressed cytokines. GO results indicated that apoptotic process, wound healing, response to oxidative stress, immune response, inflammatory response, cell proliferation, and cytokine-mediated signaling pathway were differentially expressed (Fig 6C). KEGG enrichment analysis indicated that Ras signaling pathway, TNF signaling pathway, apoptosis, PI3K-Akt signaling pathway, JAK-STAT signaling, and cytokine–cytokine receptor interaction were differentially activated (Fig 6D). There results were also consistent with our former analysis and experimental validation.

We also constructed PPI networks and calculated their priority ranks according to the BC method (Fig 6E), the MCC method (Fig 6F), and the machine learning–based random forest algorithm (Fig 6G). The intersection of the top 30 ranked cytokines calculated by these three algorithms yielded three possible crucial cytokines (Fig 6H). *Akt2*, *ApoA1*, and *ApoE* levels were significantly higher in young recipients serum than those in adult recipients (Fig 6I–J), and qPCR results also consistent with that in cytokine array (Fig 6K). Moreover, differentially expressed cytokines were closely correlated (S1H Fig), indicating that cytokines interact and cooperate in a youthful systemic environment rather than one or two cytokines dominating the biological processes.

## 3.7 Differentially expressed genes (DEGs) were associated with serum cytokines

We conducted a correlation analysis and protein–protein docking to identify the relationship between hub genes and serum cytokines. The correlation heatmap revealed that these three genes were negatively associated with serum cytokines (Fig 7A). SOCS3 could be perfectly docked with AKT2 (Fig 7B and C), indicating that DEGs were associated with serum cytokines.

## 4 Discussion

Kidney disease has an important impact on global human health, for one thing, as a direct cause of global morbidity and mortality, and for another, as a vital risk factor for cardiovascular disease [1]. CKD is a global public health threat, and has a prevalence of 9.1% that has been still increasing [1,2]. ESRD is the terminal stage of CKD and needs intermittent dialysis or kidney transplantation. Although kidney transplantation restores urinary function, transplanted patients still suffer from AKI, acute and chronic rejection, and other complications. AKI is the prior and base of rejection and other complications in some way. Thus, how to prevent and attenuate AKI is a challenging but promising approach.

The impact of a youthful systemic milieu on tissue rejuvenation, regeneration, and repair has been addressed in a number of ways, including the parabiosis model. Ludwig *et al.* successfully modified the parabiosis model, in which older rodents were physically connected to younger rodents to establish collateral blood circulation. This connection extended the life span of older rodents by 4–5 months in 1972 [34]. Because parabiosis had only limited fluid exchange among surgically connected pairs, we established a solid organ transplantation model to explore further the functional molecular

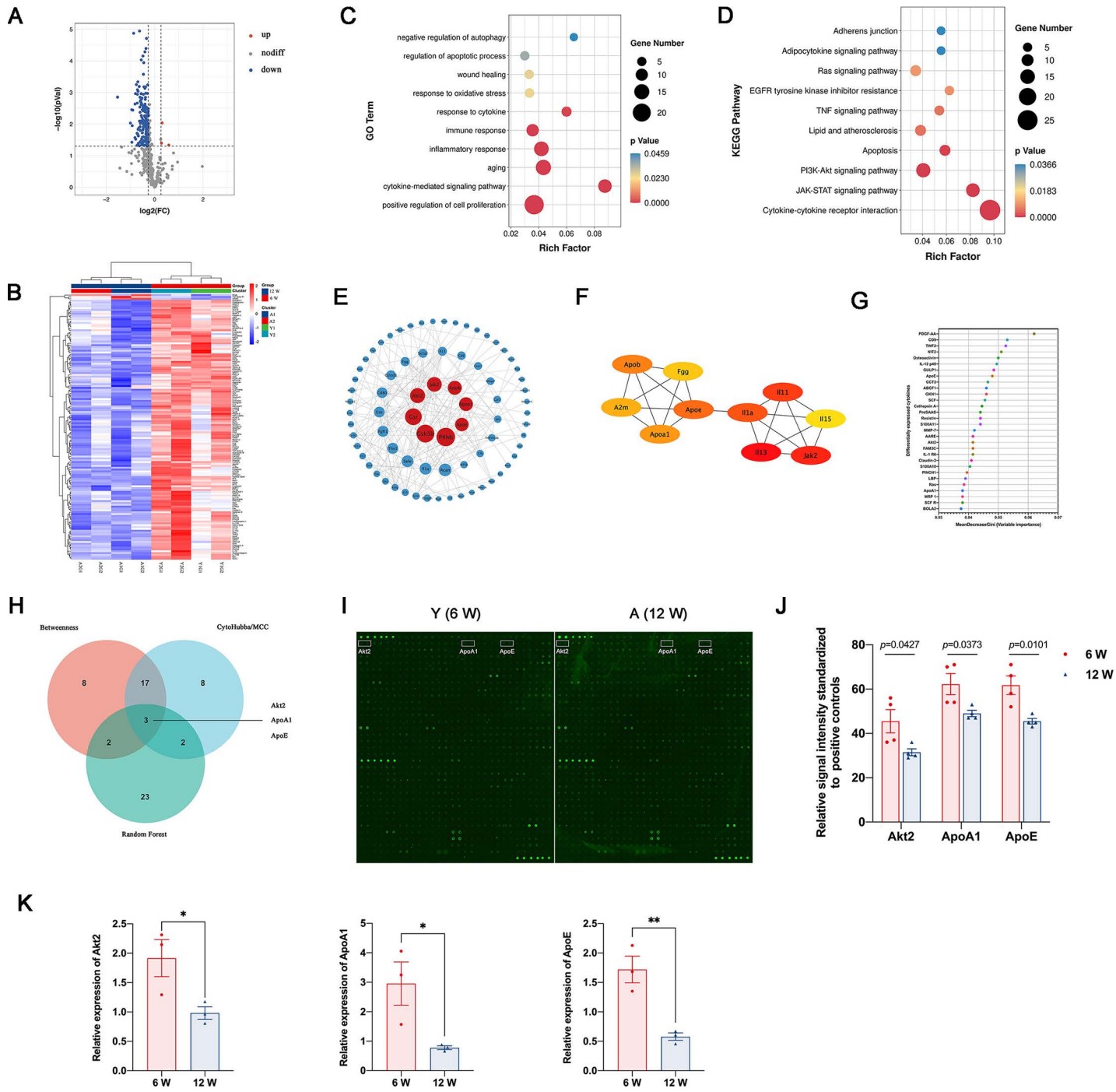

**Fig 6. Cytokine profiles were different among young and adult recipients.** (A) The volcano plot and (B) correlation heatmap of differentially expressed cytokines. (C) GO and (D) KEGG enrichment analyses of differentially expressed cytokines in Y and A group. Network construction of differentially expressed cytokines according to the Betweenness Centrality method (E) and Maximal Clique Centrality method (F). (G) Identification of top 30 cytokines based on the random forest method. (H) Venn diagram of intersection of top 30 cytokines calculated by Betweenness Centrality, Maximal Clique Centrality, and random forest algorithm. (I) Cytokine profiles of serum from young and adult recipients after kidney transplantation were compared by cytokine arrays. White boxes denote Akt2, ApoA1, and ApoE expression levels. (J) Comparison of relative expression levels of Akt2, ApoA1, and ApoE standardized to normal controls. (K) Relative expression of Akt2, ApoA1, and ApoE by qPCR.

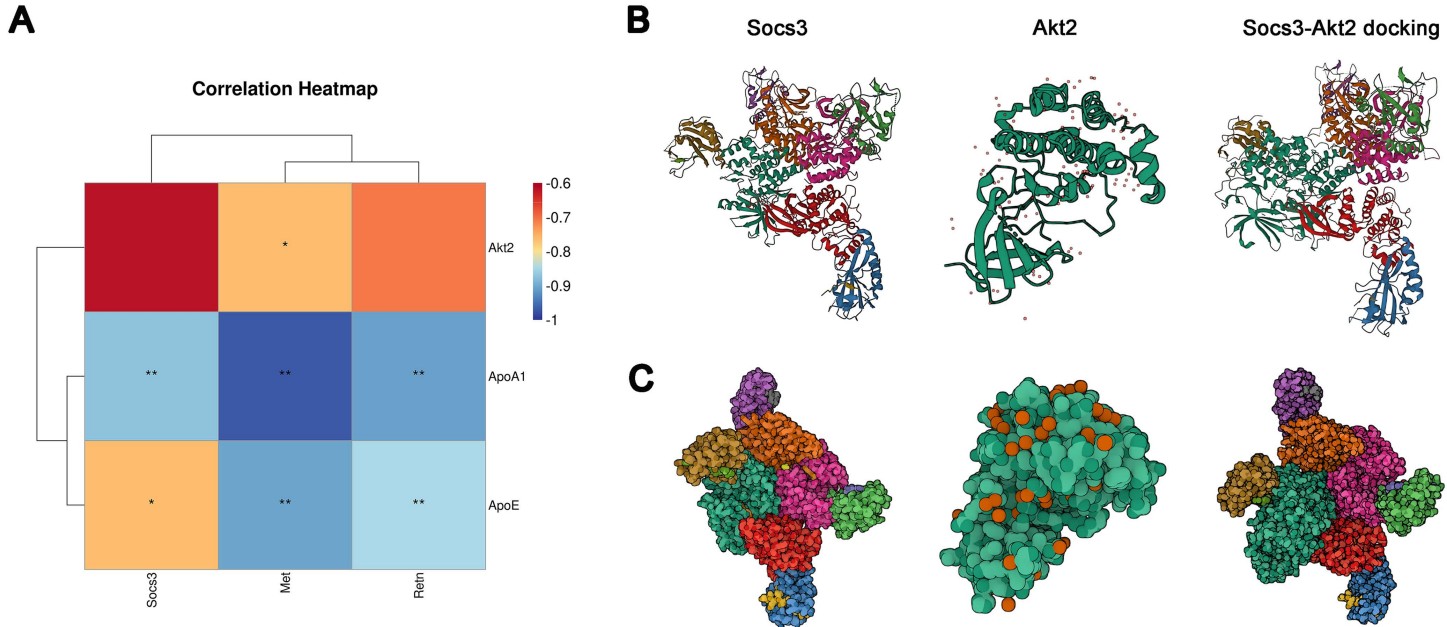

**Fig 7. DEGs were associated with serum cytokines.** (A) Correlation heatmap of identified hub genes and serum cytokines. The 3D structure of Socs3, Akt2, and the protein-protein docking (B), and its ball-shape profile (C).

changes due to a youthful systemic environment and the possible underlying mechanism. Young (8–10 weeks) and very old (19–26 weeks) rodents have been routinely used for parabiotic connection to study the influence of youthful systemic milieu on old tissues [8,11,15]. However, because very old organs are out of consideration for clinical solid organ transplantation, we did not select the very old rats as recipients. Since rats reach sexual maturity at 6–8 weeks and physical maturity at 8–10 weeks, we chose 6- and 12-week-old rats as recipients to mimic pediatric and adult KTx, respectively, to identify whether the younger internal environment would ameliorate allograft injury. However, there were still exist limitations to be considered: we did not arrange a relatively older recipients (like 12 months or even 18 months) group and didn't arrange younger kidney donors transplanted to young and adult recipients.

Surprisingly, 24 h after KTx, 12-week-old adult recipients had higher levels of AKI and decreased renal function compared with 6-week-old young recipients after receiving similar donor kidneys. Based on multi-omics, machine learning analyses, and experimental validations, we observed that the young recipients' younger systemic internal milieu alleviated kidney histological damage, oxidative stress injury, and apoptosis, and promoted repair. These results indicated that a relatively older (even in a strong stage) internal environment had increased vulnerability to injury and extrinsic stress, impaired resistance and repair potential, or both. The literature also supported that oxidative stress, inflammation, and kidney apoptosis increased with age [35,36].

It has been demonstrated that stimulating and regulating factors from a young systemic environment activated the rejuvenation and regeneration potential of old organs, rather than the migration of stem cells from young blood fluid to old tissues worked [8,12], but the functional regulating micromolecules were not well studied. Based on transcriptomics techniques, enrichment analyses of DEGs and immune-related genes indicated that cytokines and cytokine-mediated pathways were involved in AKI after kidney transplantation (Fig 2C and D; Fig 4H and I). Thus, we further conducted the proteomics analysis with a cytokine microarray, which tested 500 types of cytokines in recipient serums. About 28.6% of cytokines in young and adult recipients were differentially expressed, most of which (98%) were highly expressed in young recipients, indicating that cytokines played a vital role in multitudinous biological processes after KTx.

DEGs between young and adult recipient kidneys were associated with oxidative stress injury and apoptosis processes (Fig 2C; Fig 5I and K). Cytokines were also closely related with oxidative stress injury and apoptosis (Fig 6C and D), and most importantly, gene expression profiles were associated with cytokines as well (Fig 7A and B). Collectively, cytokines may affect gene expression profiles and participate in AKI after allogenic kidney transplantation. Cytokines play a vital role in many biological processes, including immune response, inflammation, apoptosis, oxidative stress, cell proliferation, survival, and death [37]. Kidney transplantation is an immune-accompanying process, in some way involving acute and chronic allograft rejection associated with AKI. Thus, we explored whether the immune response participated in AKI. As expected, cytokines affected immune cell infiltration, B cell and T cell activation, proliferation, and differentiation even within 24 h after allogenic KTx. Taken together, higher levels of cytokines in young recipients may alleviate kidney oxidative stress damage and apoptosis, protect kidney function, and promote injury repair via regulating the immune response. Higher levels of cytokines in young environment fluid serving as therapeutic or auxiliary agents deserve further experimental evaluation and clinical attempts.

Cardiovascular mortality and morbidity are major problems in patients with ESRD [38]. Islet transplantation combined with kidney transplantation showed protective effect in cardiovascular function [38]. In addition, successful islet-kidney transplantation demonstrated improved patient survival, long-term cardiovascular and endothelial function in type 1 diabetic kidney-transplanted patients [39]. The possible benefits of combination of cytokines and islet transplantation in KTx deserve further investigation in the future.

Regulatory cell therapy has been proven achievable and safe in living donor kidney transplant recipients [40]. Studies from seven non-randomized, single-arm phage 1/2A trials found that plasmacytoid dendritic cell levels were much higher in healthy controls than ESRD patients and higher in kidney transplanted patients 60 weeks after receiving regulatory cell therapy compared with pre-transplantation [40]. Interestingly, in our study, we also found young recipient kidneys had higher levels of both resting and activated dendritic cell infiltration (Fig 4D), indicating that dendritic cells may play a vital role in regulating immune response and alleviating kidney injury and rejection after transplantation.

Regulatory genes and cytokines have various biological functions and effects, and they also interact. Research has demonstrated that oxidative stress, apoptosis, and inflammation increase during AKI in older kidneys, and these mechanisms also interact [41]. Thus, the identification of hub regulatory genes and serum cytokines has its inevitable limitations, but it facilitates the exploration of the underlying mechanisms. The Suppressor of cytokine signaling (SOCS) family was found to play pivotal roles in immune regulation, and SOCS3 functioned as a negative feedback loop on cytokine-mediated signaling via the JAK-STAT pathway [42]. SOCS3 could also combine with AKT2 (Fig 7B). This could explain why the observed high expression of SOCS3 in adult recipient kidneys was negatively associated with lower levels of serum AKT2 concentrations (Fig 7A). Research has also indicated that AKT2 has an important role in diabetic nephropathy via regulating antioxidant and apoptosis [43], which is consistent with our findings.

Our findings may help explain why pediatric patients with kidney transplantation have a higher patient and graft survival than adults. After receiving an allogenic donor kidney, pediatric recipients tend to exhibit lower levels of stress injury and apoptosis and a higher potential of repair via regulating the immune response. However, pediatric ESRD patients usually receive organs from pediatric donors, and whether and how much a young donor kidney contributed to the prolonged patient and graft survival was not considered. Whether the recipients have an increased susceptibility to extrinsic stress and injury along with age, how cytokines interact, and the exact mechanisms still need to be clarified.

In conclusion, young recipients' younger systemic internal milieu alleviated AKI after kidney transplantation by ameliorating oxidative stress and apoptosis via regulating the immune response. Cytokine profiles differed between young and adult recipients, and cytokines involved immune cell infiltration, T cell and B cell proliferation, and differentiation, affecting kidney injury and repair. Cytokine therapy may be a potentially useful therapeutic approach in kidney transplanted recipients to minimizing the injury of oxidative stress and apoptosis and to maximize the protection of kidney function and repair after injury. Some highly expressed cytokines in young recipients serving as therapeutic or auxiliary agents deserve further experimental evaluation and clinical considerations.

## Supporting information

**S1 Fig. (A) Relative gene expression of Socs3, Met, and Retn based on Fragments Per Kilobase of exon model per Million mapped fragments (FPKM) and statistical difference evaluation by student-t test.** Protein-protein interaction (PPI) network of Socs3 (B), Met (C), and Retn (D) from STRING database. (E) Correlation analysis of Retn and 22 types of immune cells. Single gene GSEA analyses of biological processes enrichments (F) and KEGG pathways (G) based on high- (≥ 50%) and low-expressed (< 50%) Retn. (H) Correlation heatmap of top 30 identified cytokines. (TIF)

**S1 Table. The primers of rat Akt2, ApoA1, ApoE, and GAPDH.**
(DOCX)

**S1 File. The immune-related GeneLists downloaded from the ImmPort portal.**
(TXT)

**S2 File. Initial scan data of serum cytokine microarray of young and adult recipients 24 h after kidney transplantation.**
(XLSX)

**S3 File. The original western blot images of Figs 3G and 5E.**
(DOCX)

## Author contributions

**Conceptualization:** Chengjun Yu, Yi Hua.

**Formal analysis:** Chengjun Yu, Jie Zhang.

**Funding acquisition:** Yi Hua.

**Methodology:** Jie Zhang, Jun Pei.

**Software:** Chengjun Yu, Jun Pei.

**Supervision:** Shengde Wu, Yi Hua, Guanghui Wei.

**Validation:** Shengde Wu, Guanghui Wei.

**Visualization:** Sheng Wen.

**Writing – original draft:** Chengjun Yu.

**Writing – review & editing:** Sheng Wen, Yi Hua.

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
