## [Decision Letter · Decision Letter 0]

22 Apr 2025

Dear Dr. Yu,

Thank you for submitting your manuscript to PLOS ONE. After careful consideration, we feel that it has merit but does not fully meet PLOS ONE’s publication criteria as it currently stands. Therefore, we invite you to submit a revised version of the manuscript that addresses the points raised during the review process.

We look forward to receiving your revised manuscript.

Kind regards,

Paolo Fiorina, MD, PhD

Academic Editor

PLOS ONE

Journal Requirements:

4. To comply with PLOS ONE submissions requirements, in your Methods section, please provide additional information regarding the experiments involving animals and ensure you have included details on (1) methods of sacrifice, and (2) efforts to alleviate suffering.

6. Thank you for stating the following financial disclosure:

This research was funded by Chongqing Natural Science Foundation Innovation and Development Joint Fund Project (No. CSTB2024NSCQ-LZX0072), and The Joint Medical Research Project of Chongqing Science and Technology Bureau and Health Commission (No.2024MSXM002).

Reviewers' comments:

Reviewer's Responses to Questions

**Comments to the Author**

1. Is the manuscript technically sound, and do the data support the conclusions?

Reviewer #1: Yes

Reviewer #2: Partly

Reviewer #3: Yes

2. Has the statistical analysis been performed appropriately and rigorously?

Reviewer #1: Yes

Reviewer #2: Yes

Reviewer #3: I Don't Know

3. Have the authors made all data underlying the findings in their manuscript fully available?

Reviewer #1: Yes

Reviewer #2: Yes

Reviewer #3: Yes

4. Is the manuscript presented in an intelligible fashion and written in standard English?

Reviewer #1: Yes

Reviewer #2: Yes

Reviewer #3: Yes

Reviewer #1: My suggestion is to review the English and make some changing.

Here my suggestion:

In the Ethics Statement the Authors need to add the authorization number and the Authority that approved the study.

In the introduction part the authors introduced AKI without explaining the linkage with the paragraph about KTx and the linkage with CKD. And, why all the evaluation was performed at 24h after KTx, there is any previous study that report that the evaluation in 24h is sufficient for the study?

They could also specify the acronym OPTN/SRTR. In addition, the Authors speculate about graft and patient survival after KTx, but the results are still reported in OPTN/SRTR report.

In the animals’ paragraph the Authors need to report the age of the rats even if they specify in the other paragraph. Why the authors performed a bilateral nephrectomy and replace only one kidney?

For the measurement of malondialdehyde levels (MDA), total superoxide dismutase activity (T-SOD), glutathione peroxidase activity (GSH-Px), and protein carbonyl content (PC) which type of commercial kits the Authors used?

For all the assays performed in paragraphs 2.8, 2.9 and 2.12 which typo of sample the Authors use for the assays, please specify.

For both figure 3F and H I suggest a histogram, or percentage respect to total cells, with the number of positive cells, to better evaluate the results.

Reviewer #2: In this paper the authors explore the effect of kidney transplantation from adult rats into young or adult recipients and evaluate how the recipient’s milieu influence kidney damage. While the results are interesting, I think that some points should be clearer:

1. While the difference in cytokines expression is interesting, I think that a qPCR or a western blot must be performed to confirm the expression’s difference; similar to what they did with SOC3 and MET.

2. The authors assume the age of 6 weeks as young recipients and 12 weeks as old one, is there anything in literature that can confirm this assumption?

3. The kit information used to perform some experiments should be clarified, such as the ones for the measurements of malondialdehyde levels (MDA), total superoxide dismutase activity (T-SOD), glutathione peroxidase activity (GSH-Px), and carbonyl protein content (PC). In general, I suggest better specifying the method used instead of citing previous publication.

4. The ethic statement should be filled since this work includes the use of vertebrate animals.

5. I think that there is some confusion in image citations within the text, they cite figure 5-6 with data that are in figure 4-5. So, I suggest thoroughly performing a proof reading to be sure that there are not more typos and figures wrongly cited that make it harder to follow the work.

Reviewer #3: The Authors evaluate “renal replacement dialysis therapy and kidney transplantation (KTx) are the only choices for life-supporting for ESRD patients.”

Why the Authors never evaluate the possibility to describe, and evaluate the possibility of islet transplantation and describe in the introduction or in the discussion the possible advantages or disadvantages instead a whole kidney transplantation?

The Authors report in the conclusion “Cytokine therapy may be a potentially useful therapeutic approach in kidney transplanted recipients to minimizing the injury of oxidative stress and apoptosis and to maximize the protection of kidney function and repair after injury.”

Here some paper to evaluate that evaluate the advantages of islet transplantation instead to perform a whole kidney transplantation. The Authors could discuss the advantages of islet transplantation in combination with the cytokines they proposed in the paper.

• Islet transplantation is associated with an improvement of cardiovascular function in type 1 diabetic kidney transplant patients (P. Fiorina et al. Diabetes Care 2005)

• Long-term beneficial effect of islet transplantation on diabetic macro-/microangiopathy in type 1 diabetic kidney-transplanted patients (P. Fiorina Diabetes Care 2003)

**Do you want your identity to be public for this peer review?** For information about this choice, including consent withdrawal, please see our Privacy Policy

Reviewer #1: No

Reviewer #2: No

Reviewer #3: No

---

## [Author Response · Author response to Decision Letter 1]

7 May 2025

Revision Cover letter

Dear Paolo Fiorina,

Thank you very much for giving us a precious opportunity to revise our manuscript. We are appreciated with you and the reviewers very much for positive and constructive comments and suggestions on our manuscript, which was entitled “Youthful systemic milieu in younger recipients alleviates acute kidney injury via attenuating apoptosis and oxidative stress in a rat kidney transplantation model” (PONE-D-25-10480).

We have studied the editor’s and reviewers’ comments carefully and made corresponding revisions which marked with track changes in the resubmitted paper. We have made answers according to the comments. Attached please find the revised version, which we would like to submit for your kind consideration. A point-by-point response was presented below, and the main changes were as following:

(1) All misspellings, grammatical errors, and wrong words have been checked and corrected.

(2) Addressed those details as journal requirements needed.

(3) Provided the original western blot images in supplementary materials.

(4) Added the details of animal sacrifice and measures taken to alleviate suffering.

(5) Updated the financial disclosure, in accordance with the Funding Information.

(6) Added the linkage between AKI and CKD/KTx in the Introduction part.

(7) Explained the reason for performing the bilateral nephrectomy and transplanted only one kidney.

(8) Provided the histograms of Fig. 3F and H with the percentage of positive cells respect to total cells.

(9) Performed the qPCR to confirm the expression of Akt2, ApoA1, and ApoE.

(10) Specified the methods.

(11) Discussed the possible benefits of islet transplantation in improving cardiovascular function and long-term survival in kidney transplantation.

Besides, as for the financial disclosure, we would like to update the Funding Information: this research was funded by Chongqing Natural Science Foundation Innovation and Development Joint Fund Project (No. CSTB2024NSCQ-LZX0072), Program for Youth Innovation in Future Medicine, Chongqing Medical University (W0201), and The Joint Medical Research Project of Chongqing Science and Technology Bureau and Health Commission (No.2024MSXM002). The funders had no role in study design, data collection and analysis, decision to publish, or preparation of the manuscript.

We would like to express our great appreciation to you and reviewers for comments on our paper again. Looking forward to hearing your good news soon.

Thank you and best regards.

Yours sincerely,

Yi Hua, corresponding author,

E-mail: huayi_workmail@hospital.cqmu.edu.cn

Responses to editor’s and reviewers’ comments

- Journal Requirements:

[Response] Thank you so much. We have checked the whole manuscript carefully and revised these requirements as journal needed.

2.PLOS ONE now requires that authors provide the original uncropped and unadjusted images underlying all blot or gel results reported in a submission’s figures or Supporting Information files. This policy and the journal’s other requirements for blot/gel reporting and figure preparation are described in detail at https://journals.plos.org/plosone/s/figures#loc-blot-and-gel-reporting-requirements and https://journals.plos.org/plosone/s/figures#loc-preparing-figures-from-image-files. When you submit your revised manuscript, please ensure that your figures adhere fully to these guidelines and provide the original underlying images for all blot or gel data reported in your submission.

[Response] We fully understand the journal’s requirements on the original uncropped and unadjusted images. We provided the original uncropped and unadjusted western blot images (Fig. 3G, Fig. 5E) in supplementary materials, named “The original western blot images”.

[Response] All differently expressed genes (DEGs) were mapped to GO terms in the Gene Ontology database (http://www.geneontology.org/) and visualized using the online OmicStudio tools at: https://www.omicstudio.cn/tool, as stated on Methods, and no new author-generated code was available. In addition, all RNA-sequence data has been uploaded to the GEO database (accession number GSE226680), we also provided these statements in the manuscript.

4. To comply with PLOS ONE submissions requirements, in your Methods section, please provide additional information regarding the experiments involving animals and ensure you have included details on (1) methods of sacrifice, and (2) efforts to alleviate suffering.

[Response] Thank you for you reminding. We have added the details of animal sacrifice and measures taken to alleviate suffering during the experiments in Methods 2.1/2.2, which marked with track changes in resubmitted manuscript. All animal procedures were approved by the Ethics Committee of Children’s Hospital of Chongqing Medical University (protocol code CHCMU-IACUC20220429006).

[Response] Thank you very much. We checked the “Funding Information” in manuscript and the “Financial Disclosure” in the online submission system, the “Financial Disclosure” missed the Funding “Program for Youth Innovation in Future Medicine, Chongqing Medical University (W0201)”, we also included the updated statement in the cover letter.

6. Thank you for stating the following financial disclosure:

This research was funded by Chongqing Natural Science Foundation Innovation and Development Joint Fund Project (No. CSTB2024NSCQ-LZX0072), and The Joint Medical Research Project of Chongqing Science and Technology Bureau and Health Commission (No.2024MSXM002).

[Response] We updated the Funding Information in cover letter, thank you.

[Response] We have checked and updated the reference as journal requirements needed, these was no cited papers that have been retracted.

- Reviewer #1:

My suggestion is to review the English and make some changing.

Here my suggestion:

1. In the Ethics Statement the Authors need to add the authorization number and the Authority that approved the study.

[Response] Thank you for your kind review, we have checked the whole manuscript carefully and revised these misspellings, grammatical errors, and wrong words. All animal procedures were approved by the Ethics Committee of Children’s Hospital of Chongqing Medical University (protocol code CHCMU-IACUC20220429006), we also stated this in the part of “Methods 2.1” and “Institutional review Board Statement” in the manuscript.

2. In the introduction part the authors introduced AKI without explaining the linkage with the paragraph about KTx and the linkage with CKD. And, why all the evaluation was performed at 24h after KTx, there is any previous study that report that the evaluation in 24h is sufficient for the study?

They could also specify the acronym OPTN/SRTR. In addition, the Authors speculate about graft and patient survival after KTx, but the results are still reported in OPTN/SRTR report.

[Response] Thank you for your careful review and constructive suggestions towards our manuscript. We already added the linkage between AKI and CKD/KTx in the introduction part as marked with track changes. Chronic kidney disease (CKD) is a worldwide health problem with about 690 million all-stage CKD patients and its prevalence is still increasing globally, kidney transplantation (KTx) is the optimal therapy for CKD, with higher level of body health, life quality, and social functions. However, acute kidney injury (AKI) is the most critical and early factor affecting the function and long-term survival of graft after transplantation, how to alleviate AKI after KTx is still a big problem for nephrologists and urologists. Thus, we would like to find out a novel strategy to ameliorate AKI and we conducted this study. Twenty-four hours after intervention is regarded as a reliable time point to study AKI according to literature reports[1-3], so we also tested the AKI levels 24 h after KTx, and it is sufficient for the study. In addition, we specified and added the acronym OPTN/SRTR in Introduction to make it more readable.

Reference:

1. Liu D, Lun L, Huang Q, Ning Y, Zhang Y, Wang L, et al. Youthful systemic milieu alleviates renal ischemia-reperfusion injury in elderly mice. Kidney Int. 2018;94(2):268-79. Epub 2018/06/25. doi: 10.1016/j.kint.2018.03.019. PubMed PMID: 29935950.

2. Yu C, Zhang J, Pei J, Luo J, Hong Y, Tian X, et al. IL-13 alleviates acute kidney injury and promotes regeneration via activating the JAK-STAT signaling pathway in a rat kidney transplantation model. Life Sci. 2024;341:122476. Epub 2024/02/08. doi: 10.1016/j.lfs.2024.122476. PubMed PMID: 38296190.

3. Duff S, Wettersten N, Horiuchi Y, van Veldhuisen DJ, Raturi S, Irwin R, et al. Absence of Kidney Tubular Injury in Patients With Acute Heart Failure With Acute Kidney Injury. Circ Heart Fail. 2024;17(11):e011751. Epub 2024/10/18. doi: 10.1161/circheartfailure.123.011751. PubMed PMID: 39421939; PubMed Central PMCID: PMCPMC11573103.

3. In the animals’ paragraph the Authors need to report the age of the rats even if they specify in the other paragraph. Why the authors performed a bilateral nephrectomy and replace only one kidney?

For the measurement of malondialdehyde levels (MDA), total superoxide dismutase activity (T-SOD), glutathione peroxidase activity (GSH-Px), and protein carbonyl content (PC) which type of commercial kits the Authors used?

For all the assays performed in paragraphs 2.8, 2.9 and 2.12 which typo of sample the Authors use for the assays, please specify.

[Response] We added the rats age in the animal’s paragraph as suggested. Patients with end-stage renal disease (ESRD) usually have both kidneys damaged, while kidney transplantation only requires one transplanted kidney for life-supporting in clinical practice, thus, we performed the bilateral nephrectomy and transplanted only one kidney in animal kidney transplantation models, as all researchers do. As methods 2.7 stated, kidney Malondialdehyde (MDA), total superoxide dismutase activity (T-SOD), glutathione peroxidase activity (GSH-Px) levels were tested by commercial kits purchased from Nanjing Jiancheng Biochemistry Co. (Nanjing, China); and protein carbonyl (PC) kits was purchased from Solarbio Science & Technology Co. (BC1275, Beijing, China). We also specified the sample types used in Methods 2.8, 2.9, and 2.12, Thanks a lot.

4. For both figure 3F and H I suggest a histogram, or percentage respect to total cells, with the number of positive cells, to better evaluate the results.

[Response] Thank you for your reminding, we took your advice and provided the histograms with the percentage of positive cells respect to total cells, to better evaluate the results.

- Reviewer #2:

In this paper the authors explore the effect of kidney transplantation from adult rats into young or adult recipients and evaluate how the recipient’s milieu influence kidney damage. While the results are interesting, I think that some points should be clearer:

1. While the difference in cytokines expression is interesting, I think that a qPCR or a western blot must be performed to confirm the expression’s difference; similar to what they did with SOC3 and MET.

[Response] Thank you for your kind review and positive attitudes towards our work. We performed the qPCR to confirm the expression of Akt2, ApoA1, and ApoE, which differently expressed by cytokine array. Also, we completed the corresponding part of Methods and Fig. 6. The expression of target protein was based on the formula 2−ΔΔCt, and the primers used were included in Supplementary Table 1, as:

Rat GAPDH, forward: 5'-CTGGAGAAACCTGCCAAGTATG-3', reverse: 5'-GGTGGAAGAATGGGAGTTGCT-3'.

Rat Akt2, forward: 5'-CCTTATGCTGGACAAAGATGGC-3', reverse: 5'- CCGTAGTCATTGTCCTCTAGCAC-3'.

Rat ApoA1, forward: 5'-AACAGCTAGGCCCAGTGACTCA-3', reverse: 5'- TCCTCGGCCACAACCTTTAGAT-3'.

Rat ApoE, forward: 5'-TGACGGTACTGATGGAGGACACT-3', reverse: 5'- CCAGCATGGTGTTTACCTCGTT-3'.

2. The authors assume the age of 6 weeks as young recipients and 12 weeks as old one, is there anything in literature that can confirm this assumption?

[Response] As rats reach sexual maturity at 6–8 weeks and physical maturity at 8–10 weeks, we chose 6- and 12-week-old rats as recipients to mimic pediatric/young KTx and adult KTx, respectively. In order to study the younger youthful systemic milieu in young recipients on AKI after kidney transplantation, we conducted the animal transplantation model with different age[1]: 12-week SD rat donor kidneys were heterotopically transplanted into 6-week Wistar rats (young recipients, Y group) and 12-week Wistar rats (adult recipients, A group). We also explained the choice of recipient age in Discussion part: in literature reports, young (<8 weeks) and very old (19–26 weeks) rodents have been routinely used for parabiotic connection to study the influence of youthful systemic milieu on old tissues[2,3], however, very old organs are out of consideration for clinical solid organ transplantation, so we did not select the very old rats as recipients, and instead, we chose 6- and 12-week-old rats as recipients to mimic pediatric and adult KTx, respectively, to identify whether the younger internal environment would ameliorate allograft injury. Interestingly, the choice of these age has certain innovative significance in the aspect of youthful systemic milieu.

Reference:

1. Yu C, Zhang J, Pei J, Luo J, Hong Y, Tian X, et al. IL-13 alleviates acute kidney injury and promotes regeneration via activating the JAK-STAT signaling pathway in a rat kidney transplantation model. Life Sci. 2024;341:122476. Epub 2024/02/08. doi: 10.1016/j.lfs.2024.122476. PubMed PMID: 38296190.

2. Baht GS, Silkstone D, Vi L, Nadesan P, Amani Y, Whetstone H, et al. Exposure to a youthful circulaton rejuvenates bone repair through modulation of β-catenin. Nat Commun. 2015;6:7131. Epub 2015/05/20. doi: 10.1038/ncomms8131. PubMed PMID: 25988592; PubMed Central PMCID: PMCPMC4479006.

3. Liu D, Lun L, Huang Q, Ning Y, Zhang Y, Wang L, et al. Youthful syste

---

## [Decision Letter · Decision Letter 1]

11 Aug 2025

Youthful systemic milieu in younger recipients alleviates acute kidney injury via attenuating apoptosis and oxidative stress in a rat kidney transplantation model

PONE-D-25-10480R1

Dear Dr. Yu,

We’re pleased to inform you that your manuscript has been judged scientifically suitable for publication and will be formally accepted for publication once it meets all outstanding technical requirements.

Kind regards,

Paolo Fiorina, MD, PhD

Academic Editor

PLOS ONE

Additional Editor Comments (optional):

Reviewers' comments:

Reviewer's Responses to Questions

**Comments to the Author**

Reviewer #1: All comments have been addressed

Reviewer #3: All comments have been addressed

2. Is the manuscript technically sound, and do the data support the conclusions?

Reviewer #1: Yes

Reviewer #3: Yes

3. Has the statistical analysis been performed appropriately and rigorously?

Reviewer #1: Yes

Reviewer #3: I Don't Know

4. Have the authors made all data underlying the findings in their manuscript fully available?

Reviewer #1: Yes

Reviewer #3: Yes

5. Is the manuscript presented in an intelligible fashion and written in standard English?

Reviewer #1: Yes

Reviewer #3: Yes

Reviewer #1: (No Response)

Reviewer #3: (No Response)

**Do you want your identity to be public for this peer review?** For information about this choice, including consent withdrawal, please see our Privacy Policy

Reviewer #1: No

Reviewer #3: No

---

## [Editor Report · Acceptance letter]

PONE-D-25-10480R1

PLOS ONE

Dear Dr. Yu,

I'm pleased to inform you that your manuscript has been deemed suitable for publication in PLOS ONE. Congratulations! Your manuscript is now being handed over to our production team.

Kind regards,

on behalf of

Dr. Paolo Fiorina

Academic Editor

PLOS ONE